# Climate adaptation by crop migration

Lindsey L. Sloat [1,2 ✉], Steven J. Davis [3], James S. Gerber [4], Frances C. Moore[5], Deepak K. Ray [4], Paul C. West [4] & Nathaniel D. Mueller[1,2]

Many studies have estimated the adverse effects of climate change on crop yields, however, this literature almost universally assumes a constant geographic distribution of crops in the future. Movement of growing areas to limit exposure to adverse climate conditions has been discussed as a theoretical adaptive response but has not previously been quantified or demonstrated at a global scale. Here, we assess how changes in rainfed crop area have already mediated growing season temperature trends for rainfed maize, wheat, rice, and soybean using spatially-explicit climate and crop area data from 1973 to 2012. Our results suggest that the most damaging impacts of warming on rainfed maize, wheat, and rice have been substantially moderated by the migration of these crops over time and the expansion of irrigation. However, continued migration may incur substantial environmental costs and will depend on socio-economic and political factors in addition to land suitability and climate.

[1] Department of Ecosystem Science and Sustainability, Colorado State University, Fort Collins, CO 80523, USA. [2] Department of Soil and Crop Sciences, Colorado State University, Fort Collins, CO 80523, USA. [3] Department of Earth System Science, University of California, Irvine, Irvine, CA 92697, USA. [4] Institute on the Environment, University of Minnesota, St. Paul, MN 55108, USA. [5] Department of Environmental Science and Policy, University of California, Davis, Davis, CA 95616, USA. ✉email: lindsey.sloat@colostate.edu

Climate change is predicted to impact crop yields and shift areas of global cropland suitability[1–10], with potentially important impacts on land use change, biodiversity, socio-economic circumstances, and agricultural productivity. Future increases in temperature may open up new agriculturally suitable areas[1], and crops in some locations will benefit from increases in temperature[9]. However, yield responses to temperature generally increase up to a point past which they decrease rapidly[7,10,11], and on average across the globe, temperature increases are expected to have a damaging impact in the absence of compensatory management responses[4,11–13]. Yet agricultural systems will inevitably respond to these changing conditions and therefore actual losses will thus depend on the efficacy of adaptive responses by farmers[6,14–17].

Adaptation refers to actions that mitigate damages or exploit beneficial opportunities[18,19]. As a clarifying point, adaptation, as used here, refers exclusively to actions taken by humans. Some of those actions include leveraging the evolutionary processes that fit organisms to their environments (e.g., selective breeding), but the use of the word adaptation here should not be confused with evolutionary adaptation. Climate change adaptation in agricultural systems may entail changes in agronomic practices or cultivar selection that allow the successful cultivation of crops in changed environmental conditions. Herein, we refer to this class of responses as in situ adaptation. This is contrasted with changes in the geographical distribution of crops that are the aggregated result of individual decisions about crop choice, irrigation use, expansion, and abandonment. We refer to this class of responses as crop migration. Irrigation plays a special role in this distinction because the addition of irrigation to previously rainfed crop areas alters the global geographic distribution of rainfed crops. Therefore, the expansion of irrigation can be an important driver of rainfed crop migration. It is important to note that while climate is a central determinant of cropland geography[20], many political, demographic, and economic factors influence observed patterns, and therefore the extent of adaptation will be influenced by societal circumstances.

Between these two modes of agricultural adaptation, in situ responses have received much more attention, including retrospective analyses of crop temperature sensitivity[21,22], planting dates[23], cultivar selection[24], and irrigation use[25], forward-looking modeling of these responses[17,26,27], and agronomic research efforts to identify or develop more drought- and heat-tolerant cultivars[28]. Although assisted and unassisted shifts to the geographic ranges of plant and animal species has been a major topic of ecological research[29–35], there has been relatively little research on the role of migration of crop cultivation with climate changes in either the past or future, with the handful of prior studies focused on specific regions and crops[36–38].

Here, we assess historical changes in the global distribution of rainfed maize, wheat, rice, and soybean, and the growing season temperatures the crops have experienced, focusing especially on their exposure to heat. We analyze trends in growing season temperatures over the 40 years from 1973 to 2012 weighted by harvested areas using quantile regression. This approach allows us to assess trends in the warm boundary of each crop's range (which we define as the 95th percentile). We focus on rainfed crops as they are highly sensitive to temperature variability and extremes[7,21,39,40]. We find that although average growing season temperatures over areas under cultivation have increased by 0.7–1.1 °C, there has been less or no increase ($-1.6$–0.5 °C) in the upper bound (95th percentile) of temperatures experienced by maize, wheat, and rice crops because crop areas have shifted over time. In contrast, substantial breeding and agronomic investments have allowed soybeans to expand into warmer, tropical areas[24,41,42].

## Results

**Conceptual framework.** Fig. 1 illustrates the concept of adaptive migration. As temperatures change over time (from $t_1$ to $t_2$), crop areas may or may not shift. We compare the changes in growing season temperatures experienced in actual harvested areas to a counterfactual in which harvested areas are held constant at the beginning of the time period (a 5-year average from 1973–1977; Fig. 1b). Since the warmest 5% of rainfed areas exhibit substantially lower yields (by 45% on average across crops) than areas of intermediate temperatures for all crops (weighted means $t$-test, $P < 0.05$, Data Fig. 1; cf. ref. [43]), we focus our analysis on trends in the warm bounds of each crops' growing season temperatures. Analysis of the cool bounds (5th percentile growing season temperature) can be found in Supplementary Note 1 of the Supplementary Information.

No significant difference between trends in experienced and counterfactual temperatures (Fig. 1c, f) would indicate a lack of adaptive migration. However, experienced temperatures that are instead less than counterfactual temperatures would provide evidence that crop areas have shifted away from warmer (less preferable) areas or towards cooler (more preferable) areas (i.e., adaptive migration, Fig. 1d, g). Finally, experienced temperatures that exceed the counterfactual suggest that the thermal niche of the crop has expanded into warmer conditions than where it was initially grown (Fig. 1e, h). That is to say that the upper temperature limit has essentially expanded over time, assuming that the previous upper temperature boundary represents the former temperature limitation. This may be the case, for example, if new crop varieties allowed for expansion into hotter areas.

**Global trends in growing season temperatures.** Maize, wheat, rice, and soybeans exhibit substantially different growing season temperatures (Fig. 2a–d) as a result of crop seasonality and spatial distributions, but all crops experienced considerable warming during the period 1973–2012 (Fig. 2e–h). Over the 40 years, the growing season temperatures across crop areas increased by an average of 0.9, 1.1, 0.7, and 0.7 °C, respectively, (linear regression of growing season average temperatures across the top 98% of temporally-averaged harvested areas). Such warming was statistically significant over 83%, 100%, 92%, and 68% of rainfed maize, wheat, rice, and soybean areas, respectively, and none of the crop areas saw significant cooling during the period (linear regression of growing season average temperatures over time by grid cell ($P < 0.1$); Supplementary Fig. 1 and Supplementary Table 1).

**Global trends in rainfed harvested areas.** To understand how changes in crop areas may have moderated exposure to warming temperatures, we map trends in rainfed harvested areas of each crop between 1973 and 2012, with increases shown in green and decreases in brown (Fig. 3). Total rainfed and irrigated areas together increased to varying degrees for each crop over this time period (+35% maize, +0.3% wheat, +13% rice, and 159% soybean); however, because adding irrigation decreases rainfed areas, total rainfed areas for wheat and rice decreased by 10 and 7%, respectively. Rainfed maize areas increased by 24% (compared to the 35% increase in total area), and rainfed soybean areas increased by 158% (the majority of increases in soybean areas were rainfed). For reference, maps of changes to irrigated harvested area for each crop are presented in Supplementary Fig. 2. There are striking regional patterns of rainfed harvested-area change, reflecting a mix of extensification in new and existing cropping regions, crop switching, and changes in irrigation. Notable regional patterns of change include the northwesterly shift of maize and soybean areas in North America (shown by colors and centroid trends in Fig. 3e, h), the northerly shift of

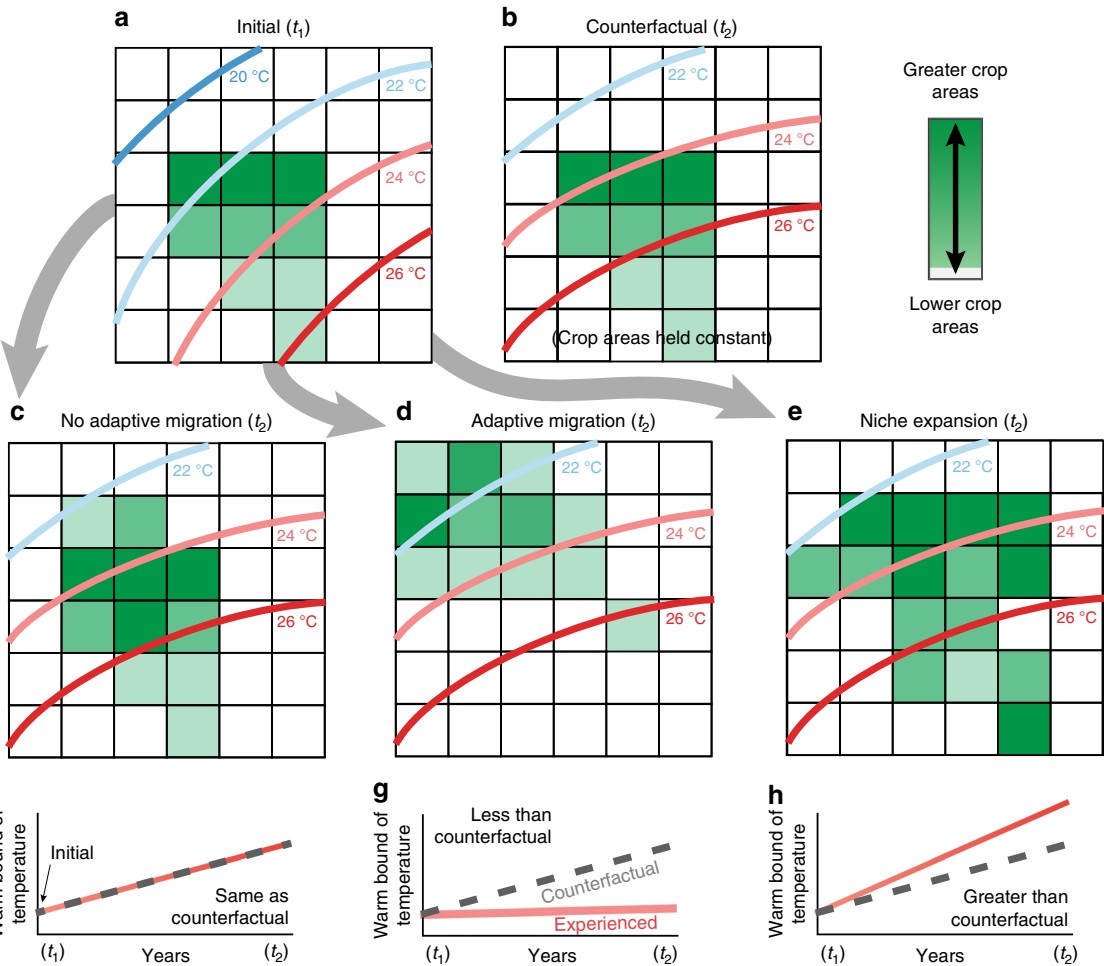

**Fig. 1 Modes of agricultural adaptation.** (**a–e**) represent theoretical gridded maps of crop harvested area. Dark green grid cells have the largest fraction of harvested area, decreasing as the shade gets lighter. Map (**a**) represents the initial time period ($t_1$), while maps (**b–e**) represent theoretical scenarios at a later time ($t_2$). As temperatures change (contours), the geographical distribution of harvested areas may or may not shift relative to **a**, the initial distribution. We compare observed changes in growing season temperatures of harvested areas to a counterfactual in which harvested areas remain constant (**b**). For the warm bound (95th percentile), no significant difference between the experienced temperature trend and the counterfactual temperature trend (**c**, **f**) would indicate no adaptive migration in response to warming; an experienced temperature trend that is significantly less than the counterfactual temperature trend suggests adaptation by crop migration to cooler areas (**d**, **g**); and an experienced temperature trend greater than the counterfactual temperature trend would suggest that crops are not only coping with temperature changes but expanding into even warmer areas (**e**, **h**).

wheat in eastern Europe (Fig. 3f), and decreased rice areas in central and southeastern China (Fig. 3c, g). It is not possible to determine exactly if one crop is being replaced with another, in part because we lack data on growing areas beyond the four major field crops presented here. For example, the contraction of wheat in Canada and Russia may be linked to the expansion of rapeseed production[9,44]; however, we are unable to show that directly. We do provide categorical maps of the largest areas of expansion and contraction among the four crops analyzed here in Supplementary Fig. 3.

**Quantile regression results**. Fig. 4 shows the trends in the warm bound (95th percentile) of crop-specific growing season temperatures that each crop experienced from 1973 to 2012 (solid red lines) as well as under the counterfactual scenario in which crop areas are maintained at their average 1973–1977 (dashed gray lines). The 95th percentile temperatures experienced by maize, wheat, and rice crops are significantly less than the counterfactual (t-test of slopes, $P < 0.05$), consistent with an adaptive migration of these crops into relatively cooler areas (Fig. 1g). In the case of

wheat, the migration has even led to an overall decrease in experienced temperatures over time (Fig. 4b). Specifically, although 95th percentile temperatures in the counterfactual scenarios for maize, wheat and rice increased by 0.68, 1.01, and 0.67 °C, respectively, the increases in 95th percentile temperatures actually experienced by these crops are much smaller, at 0.35, −1.57, and 0.46 °C (0.34, 2.58, 0.20 °C less), respectively. In contrast, the warm bound experienced by soybeans (+1.48 °C) was greater than the counterfactual (+0.77 °C; Fig. 4d), consistent with thermal niche expansion (Fig. 3h). All quantile regression model results are presented in Supplementary Table 2. Temperature changes derived from quantile regression model slopes as well as P-values for the test of slope differences between counterfactual and experienced models are presented in Supplementary Table 3. The supplemental material also contains results for other upper boundary percentiles, including the 90th, 93rd, 97th, and 99th percentiles, which do not differ in direction or overall interpretation from the 95th percentile.

The supplemental material includes additional information and analyses on lower bound (5th percentile) temperature changes. The interpretation of these results is more nuanced because it is

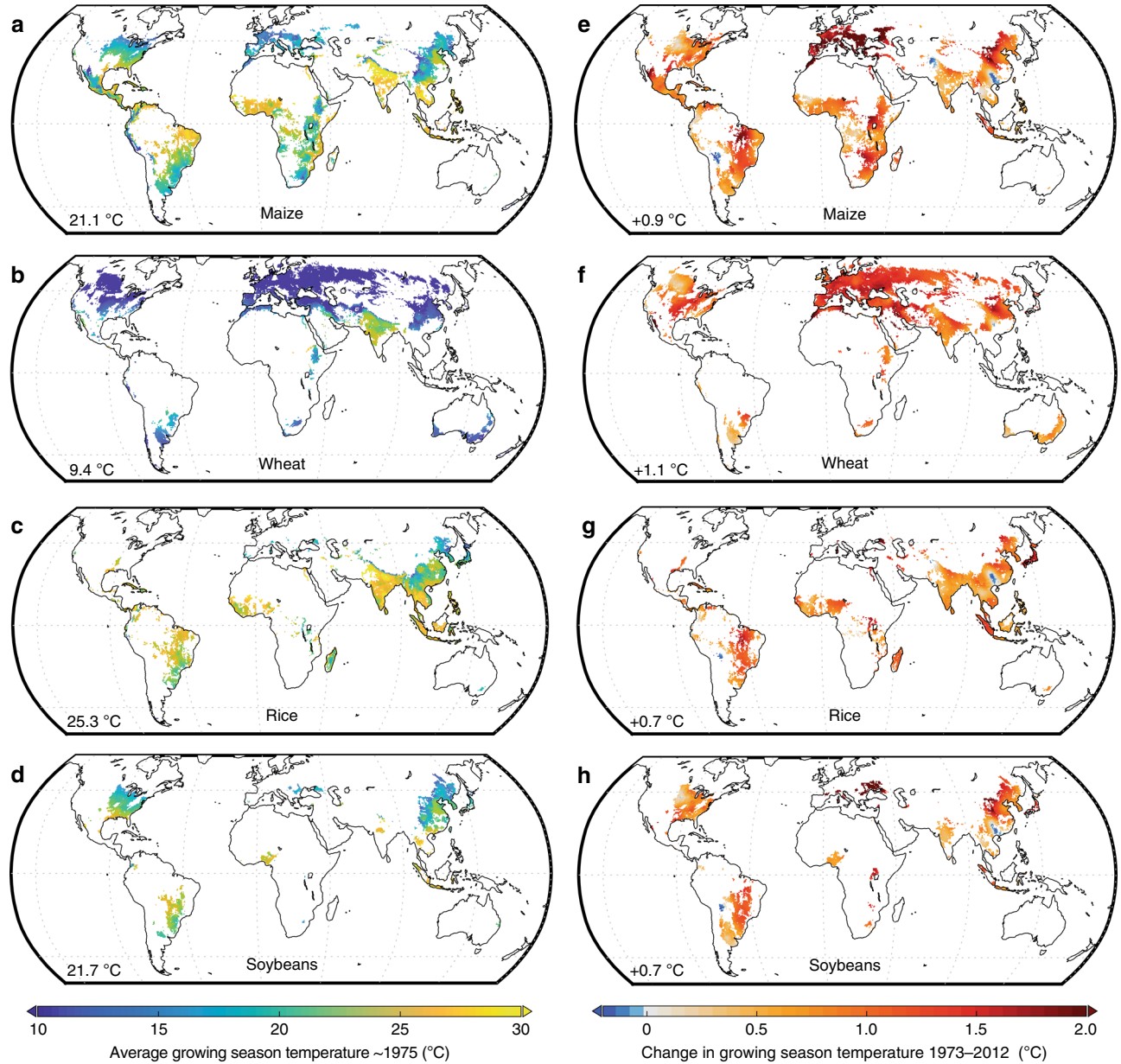

**Fig. 2 Growing season temperatures and temperature trends.** Average growing season temperatures (**a–d**) have increased from 1973–2012 (**e–h**) across most growing areas for maize, wheat, rice, and soybeans. Average growing season temperatures are shown (left) for the beginning of the time period (circa 1975) by averaging the growing season temperatures from 1973 to 1977. Trends are calculated using linear regression, and all values are displayed across the top 98% of average rainfed harvested areas. Temperatures are averaged across multiple growing seasons (e.g., spring and winter wheat) where relevant. The lower left corner of each map shows the globally averaged temperature (left) or change (right), weighted by average rainfed harvested areas.

not entirely clear if moving into warmer areas or even historically cooler areas than the counterfactual is preferable, and when crops do move into even cooler areas than the counterfactual they are often still experiencing warmer temperatures than the beginning of the time period due to climate change. In addition, it is not clear that cold temperatures are as serious a constraint on production as hot temperatures[7], meaning the adaptive response to hotter temperatures would be expected to be less pronounced. In our dataset, maize, wheat, and soybean growing in the coldest 5% of their range have lower yields (by about 16% on average) than the middle 90% of areas, but rice actually has yields that are about 6% higher in the coldest area (Supplementary Fig. 5). The magnitude of change between the counterfactual and observed models was relatively small for all crops. We found that maize, rice, and

soybean experienced 5th percentile growing season temperature increases that were 0.14, 0.49, and 0.22 °C less than they would have been in a counterfactual situation, while wheat experienced increases in growing season temperature that were 0.23 °C greater than the counterfactual.

## Discussion

Crop migration has mediated crop growing season temperatures. As shown in previous studies[45], rainfed crops are currently experiencing average growing season temperatures that are hotter than they were in the 1970s because of climate change (Fig. 2e–h). However, harvested-area changes have modified the degree of extreme temperature exposure. The 95th percentile temperature

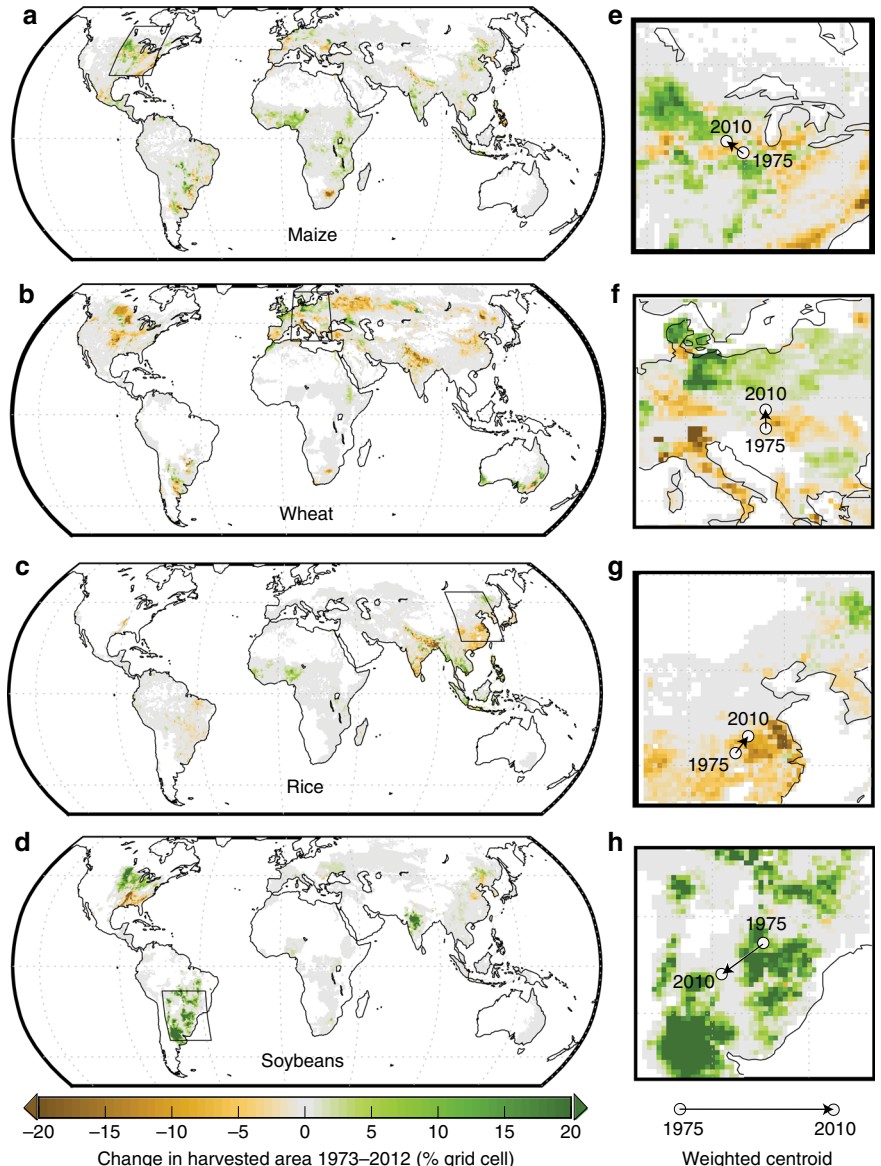

**Fig. 3 Trends in rainfed harvested area between 1973–2012.** Trends are calculated for maize (**a**), wheat (**b**), rice (**c**), and soybeans (**d**), using linear regression, and all values are displayed across the top 98% of harvested areas. Locations in brown/orange are experiencing decreases in rainfed harvested areas, locations in green are experiencing increases, and locations in gray are experiencing near-zero trends. Insets highlight areas that have experienced large changes and illustrate global trends, including a northwestern shift of maize (**e**) in the US, a northward shift of wheat in Europe (**f**), a northward shift in rice in China (**g**), and a general increase in soybean in Brazil and Argentina (**h**). White circles indicate the harvested-area weighted centroid of that region circa 1975 and the black arrowhead indicates the weighted centroid of that region circa 2010.

of rainfed wheat is now actually cooler than it was in ~1975. Wheat (which includes winter wheat varieties) would have experienced the largest upper boundary temperature increase under the counterfactual situation but instead saw the largest decrease due to substantial movement of harvested area and expansion of irrigation. The warmest wheat growing areas, mostly in South Asia, expanded irrigation over this time period[46]. Additionally, wheat shifted out of some of the coldest areas of Canada and Russia. The result is that rainfed wheat is growing in overall more favorable temperatures than would have been the case without these changes to distribution and irrigation.

Maize production in North America has likely benefited from a shift away from the American Southeast towards the upper-Midwest, where farmers are planting varieties that take advantage of longer growing seasons and less frequent extreme heat[23,47,48].

Rainfed rice is moving into slightly cooler environments. The avoidance of high temperature exposure may be driven by increase in rice irrigation in the warmest parts of its range, including Brazil, Spain, and India, as well as a northward migration in some areas, including China. Wang and Hijmans[37] have reported climate adaptation in Chinese rice via a northward geographic expansion since 1949, ameliorating warming trends and leading to a small overall benefit to national yields.

Unlike the other crops, rainfed soybean is expanding its upper thermal temperature niche while experiencing an increase in the lower end of the temperature distribution due to a warming climate. Over the time series studied, soybean harvested areas expanded by 158%, much more than the three other crops, primarily in warm, tropical areas such as India and Brazil where expansion has been assisted by the development of new varieties[24,41,42]. Consequently,

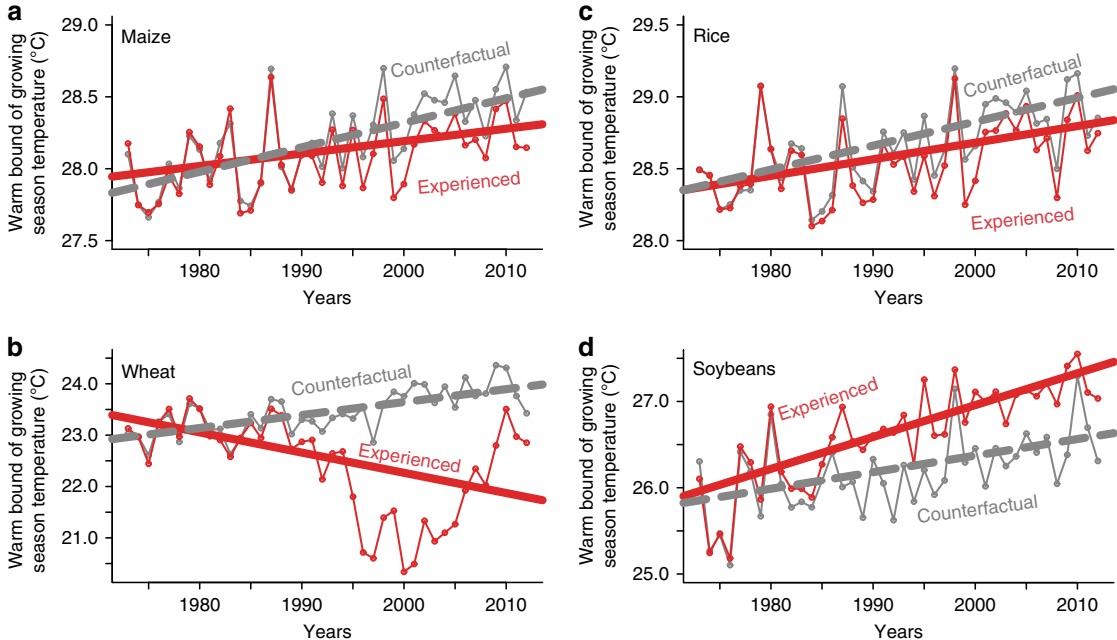

**Fig. 4 95th percentile trends in growing season temperature over time.** Plots show the quantile regression trends in the warm bound (95th percentile) of growing season temperatures between 1973–2012 for rainfed harvested areas. Results are consistent with climate adaptation for maize (**a**), wheat (**b**), and rice (**c**), and niche expansion for soybeans (**d**). Red lines indicate observed temperature trends that are influenced by changes in crop area and climate, whereas gray dashed lines represent a counterfactual scenario where rainfed harvested areas remain static at the 1973–1977 distribution.

of all four crops, soybean experienced the largest observed increase in exposure to extreme warm temperatures. Changes in rainfed harvested areas and growing season temperatures are summarized by region of the world in Supplementary Table 5.

The scope of analyses presented here covers a single, but important, climate variable (average growing season temperature), and does not address the social and economic factors that influence crop locations. Future work on the role of crop migration in climate adaptation should focus on other climate and edaphic factors, beyond temperature, that play an important role in crop production. The addition of dynamic planting and harvesting dates could provide important, additional information on how in situ adaptations of planting dates or cultivar choice have mediated crop exposure to climate. While we have shown that global maize, wheat, and rice rainfed crops have moved towards areas with temperatures that are generally considered more favorable, the direct effects on yields were not considered. This was, in part, due to a lack of temporally dynamic global yield data at high spatial resolution for only rainfed crops for this time period. Future studies relating climate change, crop distributions, and yield would improve our understanding of adaptation strategies that could aid crop production. Additionally, our study does not attempt to disentangle the relative influences of prices, trade, offshoring of agricultural production, access to markets and market changes on crop area distributions.

It is important to note that adaptive crop migrations depend on the continued ability to move growing areas and expand irrigation, and the long-term prospects for continued use of these practices are not clear. As the climate changes, areas of crop suitability may shift to the particular detriment of developing countries that tend to be warmer, and further work is needed to ascertain the possibility for adaptive migration as well as the ecological and geopolitical implications of migrating food production. Shifting crop areas may not be a sustainable method of adaptation for many reasons. Expanding agriculture into new areas is extremely environmentally damaging, decreasing carbon storage, harming

water quality, reducing wildlife habitat, and biodiversity[49–52]. For example, the expansion of soybean production in South America has had damaging consequences for the highly biodiverse Cerrado biome[53]. The majority of harvested-area changes in our dataset appear to result from crop switching (Supplementary Fig. 6) or changes to irrigation (Supplementary Fig. 2); however, crop switching can be limited when new technologies or methods are required. Further, increasing irrigation can be problematic as an adaptation strategy when measured against the impact on available water resources or the effect of increased runoff on water quality[54], and the long-term sustainability of irrigation expansion is not clear given existing stress on water supplies[55–57].

Despite these limitations, our results show that crop migrations have already mitigated high temperature exposure for the world's most important cereal crops. These changes in crop area may be as important as more commonly considered in situ adaptation strategies when investigating climate change impacts on agriculture.

## Methods
**Data preparation**. Average growing season temperatures are calculated by summing average daily temperature values for each day of the growing season and dividing by the length of the growing season. Growing seasons are defined differently for each crop using global, gridded maps of crop-specific planting and harvesting dates (day of year) from Sacks et al. (ref. [58]), provided at the 5 arc minute resolution. These maps do not vary through time, and thus growing season lengths are constant in this analysis. Temperature data are provided at 30 arc minute (half degree) resolution, so growing season data are upscaled from 5 to 30-min. Because temperature data are monthly, a linear interpolation is applied to calculate daily values before trimming to the growing season length and determining the mean daily growing season temperature (referred to as average growing season temperature). We compare our results to the CPC Global Temperature data that are available at a daily timescale, but are only available starting in 1979. The results found using this dataset are not substantially different from those using the CRU data. Results found using the CPC Global Temperature data are found in the supplemental material in Supplementary Tables 6 and 7.

The fraction of irrigated area in each grid cell is found by scaling the fraction of irrigated area (from MIRCA2000[59], as summarized in Mueller et al.[60]) as a maximum proportion of crop area in each grid cell. A time series of crop-specific

irrigation fraction was developed using historical ratios of area equipped for irrigation from Siebert and Döll (ref. [61]) (version "AEI_EARTHSTAT_CP") and a linear extrapolation beyond 2005 constrained between 0 and 1 (inclusive).

Global, gridded harvested-area data for maize, wheat, rice, and soybeans are from Ray et al. (ref. [9]) for the years 1973–2012. Data are drawn from over 20,000 administrative units and are provided on a 5 arc minute resolution grid. We upscale these data to 30 arc minute resolution for consistency with the temperature data.

Rainfed harvested areas were calculated by multiplying the total crop-specific harvested area of each 30-minute grid cell by the fraction of that grid cell that is rainfed, where the rainfed fraction is the additive inverse of the irrigated fraction. Dynamic harvested areas (used in the observed model) are calculated using harvested areas and rainfed fractions that change annually from 1973 to 2012. Static harvested areas (used in the counterfactual model) are calculated using harvested areas and rainfed fractions from the beginning of the time series, a 5-year average from 1973 to 1977.

**Quantile regression analysis.** Quantile regressions of average growing season temperature over time were analyzed at the global level. The time series spans 40 years, 1973–2012. The analysis was done in R using the rq function from the quantreg package[62] at various percentiles of temperatures ($\tau = 0.9$, 0.93, 0.95, 0.97, and 0.05). We used total harvested areas (in hectares per grid cell) as weights so that grid cells with more land devoted to growing that crop were weighted more heavily in the quantile regression, thus characterizing the entire distribution of temperatures experienced by each crop. Weights for the observed regression utilize dynamic rainfed harvested areas (described above). Weights for the counterfactual regression utilize static rainfed harvested areas (described above). Quantile regression model results are presented in Supplementary Table 2.

**Hypothesis tests.** The hypothesis test for adaptive migration at the warm bound of temperatures (95th percentile) (Fig. 1g) involves a statistical test of whether or not the time trend in temperatures from the observed model ($\beta_o$) is significantly less than the time trend of temperatures from the counterfactual model ($\beta_c$); cf. ref. [63]. In other words, whether or not the difference ($\Theta$) between these slope coefficients is significantly less than zero.

The observed model is the quantile regression of average growing season temperatures over time weighted by dynamic rainfed harvested areas. $\beta_o$ is the slope of this model. The counterfactual model is the quantile regression of average growing season temperatures over time weighted by static rainfed harvested areas. $\beta_c$ is the slope of this model.

$\Theta$ is found as the difference between $\beta_o$ and $\beta_c$:

$$\Theta = \beta_o - \beta_c \tag{1}$$

To determine whether adaptive migration is occurring, we bootstrap $\Theta$ 500 times and find the $P$-value as the proportion of data above 0.

The hypothesis test for niche expansion at the warm bound (Fig. 1h) involves a statistical test of whether or not the time trend of the observed model is significantly larger than the time trend of the counterfactual model. The logic of this is the same as described above for climate adaptation, except the test for niche expansion is if the difference between $\beta_o$ and $\beta_c$ ($\Theta$) is significantly greater than zero and so the $P$-value is found as the proportion of the distribution of $\Theta$ that is below 0. If $\Theta$ is not significantly different from zero, then there is no evidence for adaptive migration or niche expansion (Fig. 1f).

**Linear regressions of temperatures and harvested areas.** Changes in average growing season temperature (Fig. 2e–h) and changes in rainfed harvested area (Fig. 3) were analyzed as the changes in these variables over time at the grid cell level. Regressions were analyzed and mapped globally in Matlab. Maps of $R^2$ values and $P$-values are provided in Supplementary Fig. 1. The percentage of global rainfed harvested areas that have experienced significant increasing or decreasing trends is presented in Supplementary Table 1. All models were tested for nonlinear responses and three of eight models had significant ($P < 0.05$) second degree polynomial terms (Supplementary Table 4 and Supplementary Fig. 4). However, the inclusion of these terms did not affect the overall conclusions of the paper, so the more easily interpreted linear response terms were retained.

**Yield and temperature comparisons.** In order to confirm that rainfed yields are typically lower in the extreme ends of the growing season temperature distribution, as would be expected from a host of agronomic and climate impact analyses[21–23,43], we compared yields on areas below the 5th percentile of temperature, in the 5th–95th percentile temperature range, and above the 95th percentile of temperature using a weighted means $t$-test with bootstrapped standard errors from the weights package in R[64]. This analysis does not attempt to causally isolate the impact of temperatures on our cross-sectional rainfed yield data, but rather to confirm broad patterns. Subnational data on yields for rainfed and irrigated crops circa 2000 (1998–2002) are from Siebert and Döll[61], who utilized high-resolution global yield datasets, crop-specific irrigation data, and aridity-based regression models to disaggregate rainfed and irrigated yields. Growing season temperatures

are averaged over this same time period. Results of this comparison for rainfed crops are presented in Supplementary Fig. 5.

**Data availability**
All data are from publicly available sources. Temperature data are from the Climatic Research Unit (CRU) TS v.4.02, available at https://crudata.uea.ac.uk/cru/data/hrg/cru_ts_4.02/[65]. We chose CRU data because they are widely-used, based on observations, and available at a finer resolution[66] and longer duration[67] than other global gridded temperature datasets. We compare our results to those found using CPC Global Temperature Data and those data are provided by NOAA/OAR/ESRL Physical Science Division, Boulder, Colorado, USA, from their website:https://www.esrl.noaa.gov/psd/. Crop planting and harvesting dates are from Sacks et al. [58], available for download at this website: https://nelson.wisc.edu/sage/data-and-models/crop-calendar-dataset/index.php. Rainfed and irrigated crop yield estimates relevant to the year 2000 (1998–2002), as well as changes in area equipped for irrigation over time are from Siebert and Döll (ref. [61]); yield data are available from the author by request, while historical irrigation data are available for download at this website: https://mygeohub.org/publications/8/2. Crop-specific irrigation data are from MIRCA2000[59], available for download at this website: https://www.uni-frankfurt.de/45218031/data_download. Crop harvested areas are from public data sources as described in Ray et al. (ref. [9]).

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

## Acknowledgements

L.S. and N.M. were supported by USDA NIFA (2016-67012-27434). S.D. and F.M. were supported by the U.S. National Science Foundation and USDA NIFA (INFEWS grant EAR 1639318, NIFA grant 12225279). J.G., D.R., and P.W. were supported by the Belmont Forum/FACCE-JPI funded DEVIL project (NE/M021327/1) and the Institute on the Environment.

## Author contributions

L.S., N.M, J.G., and S.D. conceived the project; D.R., P.W., and J.G. provided data; L.S. and J.G. compiled data; L.S. led data analysis with assistance from F.M., S.D., and N.M.; L.S., S.D., J.G., F.M., D.R., P.W., and N.M. wrote the manuscript.

## Competing interests

The authors declare no competing interests.
