## [Peer Review File · Nature Communications]

Reviewers' comments:

Reviewer #1 (Remarks to the Author):

The manuscript explores the trends in growing season temperature of several crops (maize, wheat, rice and soybean) as their growing areas move into new environments in response to climate-related temperature changes (adaptive migration). These trends are evaluated against the counterfactual of no adaptive migration.

Overall, the concept of the paper is interesting, and the authors present some new and useful data. However, the current version of the manuscript is confusing for the uninitiated reader, who needs to put considerable effort into understanding the analysis and the conclusions drawn. In other words, clarity could be considerably improved. For instance, introducing several concepts within the same sentence is problematic, inviting misinterpretations and misunderstandings (e.g. Lines 17-23 introduce the concepts of two different types of adaptive responses, namely damage avoidance vs niche filling, as well as in situ adaptation vs migration). I understand the need for brevity, but as Einstein supposedly said: explanations should be as simple as possible, but not simpler!

The authors should consider re-writing the abstract by more meaningfully summarising their findings and the likely implications. The current version left me with a very strong 'so what?' question and did not entice me reading the rest of the paper.

As a general point, I would have expected the authors to be clearer in articulating that shifts in production areas might not always be in response to a changing climate. Rapid urbanisation, competing demands on land from other sectors, changes in demographics and markets can all be contributing factors for crop migration and these effects might be difficult to separate from true climate drivers. While this is acknowledged (lines 140-142 and 152-154), the point should be made earlier and more forcefully.

The analysis appears sound, although I am not an expert in statistics. I have one question, however: given that the slope of differences between counterfactual and experienced temperatures were tested using P-values, I am wondering if the authors did a check for normality of their data, which is a prerequisite for using P-values (Extended Data Figure 2, a-d).

Data presented in Fig. 4 seems inconsistent in terms of its interpretation. While I agree with the interpretation of the warm bound growing season temperatures (left panels in red), I could not understand why wheat (f) was considered consistent with adaptive migration, while maize and soybeans were not (I agree with the interpretation for rice, panel g). The interpretation appears to be based exclusively on the statistical significance tests as shown in the Extended Data Table 4), but this is not intuitively apparent by visually inspecting the slopes in Fig. 4. For me, this raises the question: Could this simply be an artefact of the statistical analysis? Would a slight change in the significance threshold result in a different interpretation? While the slopes might be statistically significantly different, the magnitude of difference might be irrelevant from a practical perspective (see several papers debating 'the insignificance of significance testing').

Specific comments

Lines 2-3 Yet the mechanisms and capacity for agricultural adaptation to climatic changes remain highly uncertain ...

The term 'uncertain' does not adequately convey the real issue surrounding agricultural adaptation. Adaptation is complex and highly context-specific. Some adaptive measures are very certain, but often they cannot be scaled. They need to be seen and evaluated within their socio-economic context. Using the term 'uncertain' is probably one of the most unhelpful terms in this context and simply provides fuel for a 'do nothing' response (the supposed 'uncertainty' of climate science is largely

responsible for the lack of policy action so far).

Lines 13-14 If migration continues into the future, it will have important implications for the magnitude and extent of agricultural climate change impacts ...

While this statement is true, it fails to acknowledge some of the potentially much larger impacts of such migration on e.g. landuse changes or socio-economic circumstances. It is also rather meaningless, because we know a priori that migration will continue to occur.

Lines 15-17 ... global yields of wheat, rice, maize, and soybeans are expected to fall by roughly 7%, 6%, 3%, and 3% for every degree Celsius increase in global mean temperature² ...

I find it hard to believe that there is a linear relationship with further temperature increases and yield reductions, even if this is the conclusion by Zhao et al (2017). I haven't checked the reference and am wondering if this is the correct interpretation of their work? I would expect that as temperatures rise, impacts would become more severe and might even cross thresholds where we could see step-changes in yield patters. I'd like to see some better justification, in addition to this single reference, for this response.

Lines 17-18 However, agricultural systems will surely respond and actual losses will thus depend on the efficacy of such 'adaptive' responses by farmers^{4,8,12} ...

This sentence seems without context. I am not sure which adaptive responses by farmers the authors refer to.

Reviewer #2 (Remarks to the Author):

This is an original contribution of broad general interest, that advances understanding of information of high importance. The approach is novel, and asks a different question than has been previously addressed: to what extent are the growing temperatures of major crops that are actually being experienced moderated by "crop migration" in the face of climate change? The approach is valid and the statistical methods used seem to me generally to be appropriate. The focus on temperature extremes is excellent and is biologically meaningful. My main comments concern the potential for misinterpretation, and also, I found parts of the ms. confusing or difficult to understand.

I am concerned that the narrow scope of the conclusions and interpretation might lead to claims (particularly by news media, public, policy makers) that "everything is fine"--crops are being moved to more favorable areas, so why worry about changes in temperatures affecting yield? First, I would like to see the caveats discussed (more than as brief remarks) that future movement of crops will be constrained in many ways--by urbanization, land ownership, water rights, national boundaries, and other land-use issues. For example, if farmers' lands can no longer sustain productivity of the crops they are growing, they can't just move over to someone else's land; there may be winners in this scenario but also losers. Limits to irrigation are mentioned, but too briefly in my view, and limitations in precipitation (another consequence of climate change) will also impede future maintenance of crop productivity and the ability to migrate crop production. Perhaps most importantly, the impacts of crop migration--current and future--on biodiversity are also staggering. Currently some huge proportion of productive terrestrial land surface is dominated by human use; if and when croplands move, they often are overtaking more and more land occupied by natural communities. The story of the expansion of soy production in South America reported here is also a little known story of the obliteration of the highly biodiverse cerrado, for instance. These issues are not mentioned here and in my view it is important to include that context for the changes reported here.

There were a number of areas I found confusing. I don't understand what thermal niche refilling means; this could be better explained. The explanations for the cool bounds were confusing (lines 57-64 and lines 100-107); these portions of the ms. were just hard to understand. The cool season boundary may be less meaningful than growing degree days (is the cool season boundary reducing

growing season length?). Since growing seasons were apparently calculated, it would be valuable to include those results for interpreting the findings.

Are wheat and rice contracting overall or just changing to irrigation(Fig 3)? Is wheat being replaced by maize, or is it just that rainfed wheat is being replaced by irrigated wheat? Why is wheat shifting out of the coldest areas in Canada and Russia? The expansion of irrigation is mentioned but could be somewhat expanded; this is an important part of the story.

Figure 1 was useful, but somewhat unclear; what are the axes? Are the boxes meant to show total area? Why is it different in c and d compared to a? What does the shading indicate?

In Fig 2 a-d does it make sense to show the average growing season temperature, if it is changing rapidly? Maybe the baseline year and the slope for change over time would make more sense?

I don't understand the point of extended figure 2 (why is this useful information? What does it tell us?); spatially explicit values of fit and significance seem, well, kind of odd. Maybe there's something there I'm missing, so please explain if so, or delete this figure. Extended data Fig 1 could be explained better; the caption and ms. text are unclear. Extended figure 3 was just hard to understand (relative change? I found this hard to interpret).

I had a couple of comments on the graphics: The colors in Fig 3 would be better using a different palette, because it is showing something very different (the colors in the other figures indicate, appropriately, temperature or temperature-related data; these maps are referring to something completely different). I found the 'shadows' on the maps unnecessary--being a fan of Edward Tufte, I would like graphical elements to convey information, not decoration, and the shadows are just a distraction from the data.

Jessica Gurevitch

Reviewer #3 (Remarks to the Author):

Review: Climate Adaptation by Crop Migration (Sloat et al. 2019)

1. Summary

In this paper, the authors focus on crop migration as a mechanism for climate change adaptation by looking at the upper (95 percentile) and lower (5 percentile) bounds of average temperature distributions. The analysis focuses on rainfed maize, wheat, rice, and soybeans through the world from 1973-2012 to study how land use for each crop has changed relative to a linear trend in average temperatures. They construct a counterfactual based on 1973-1977 land uses for each crop and compare against the entire sample. The main results suggest crop migration has mitigated most of the damage from warming as opposed to thermal tolerance of crops. These results provide a worldwide impact that extreme heat will play on the distribution of crops and have important implications for food security under climate change.

While the paper attempts to address world-wide crop migration as a mechanism for climate change adaptation the work is similar to Zhu and Tara (2018) and the data used in the analysis could be improved upon (Zhu and Tara provide a similar analysis with better data). Further, it is unclear why crop migration is occurring and the underlying mechanism is not addressed which is a major gap in the current literature. Instead, the authors discuss these possible mechanisms, such as prices, trade, access to markets as limitations to the paper and do not address them in the analysis. Understanding that these mechanisms are difficult to address, the paper does not improve our understanding of crop migration but that it might exist and may mitigate climate change impacts.

2. Major Comments

Good

- Paper is well written and the authors are explicit that their analysis only focuses on crop migration using average temperature distributions. Discussion section addresses the major limitations in the paper and is upfront about what they are doing.
- The discussion of crop migration and changes in crop area distributions sets up the understanding of the analysis nicely.
- Figures and tables are organized and labeled properly. The cropping pattern (Fig 1) is a nice complement to the climate change adaptation discussion.
- Discusses the main papers in the literature but omits one similar to their analysis (Zhu and Tara 2018).

Critic

- The main critic for this review is that the data and modeling strategy could be improved upon. Further, the analysis estimates linear trends when the majority of climate impacts are nonlinear (second-order direct and indirect effects).
 - Data:
 - Why only use average temperatures? This is a midpoint which does not account for the entire distribution of temperatures during the day or growing season. Degree days would be a better measure and likely more defensible given extreme degrees are known to be strong predictors in yield models.
 - Linear interpolation of daily average temperatures using monthly values is problematic. Instead of linear interpolation of monthly data, I would suggest using a reanalysis gridded data set. There is better data available for the analysis that would improve the confidence in the results.
 - What is the climate variable? Average daily temperatures? The main variable accounts for daily average temperatures which are a measure of weather, not climate. A better variable would be previous 30-year average temperatures or a rolling mean of temperatures under different windows (e.g. 30 years).
 - Modeling strategy:
 - Quantile regressions make sense to capture changes on the ends of the distribution, but I would suggest using nonlinear quantile regression. While the linear trends do capture average temperatures for most crops it is not always the case (see wheat). I suspect your results will change with different percentile bounds and climate variables.
 - How sensitive are the bounds? I would suggest showing how results change with different percentiles (90%, 99%, etc.)
 - The data used in the analysis is based on linear interpolation or a linear variable (average temperatures) which is then modeled using a linear model. An analysis using nonlinear assumptions would improve confidence in the results.
 - While on average temperatures are increasing what about regionally? I would suggest adding a region fixed effect to account for the fact that not all areas will increase uniformly. A break out of these regions and increases would be another figure to look at and see if there is a significant difference.

3. Minor Comments

- Line 19-23 is an important sentence but the sentence is long and clunky. I would suggest breaking it up.
- Line 36: Defend 5% and 95% percentiles.
- Suggestion: A plot of temperature distributions across yields and how the 5% and 95% percentile

distribution differ to actual vs counterfactual.

- Suggestion: a plot of how much yield is located on those upper and lower bounds. Also, look at changes in actual vs counterfactual.
- Line 57: "Changes in the cooler bound ... will make areas at this bound more favorable." Is this true for all crops? Increases in temperatures at the lower bound for wheat might be less favorable depending on location. For example, wheat varieties favor winters that rely on frost levels for production.
- Line 90: Why 1973-1977 as the counterfactual? Is this a stable time period to conduct a counterfactual with? A discussion as to why this interval was used and how sensitive the results are to changing these dates (e.g. 1973-1980).
- Line 139: are there implications for complementarities, such as corn and soybean? How has warming in soybean areas related to corn?
- Suggestion: a plot of changes in trend in different countries/regions.

References

Zhu, Xiao, and Tara J. Troy. "Agriculturally Relevant Climate Extremes and Their Trends in the World's Major Growing Regions." *Earth's Future* 6, no. 4 (2018): 656-672.

Response to Reviewers

We thank all three reviewers for their thoughtful feedback and comments. In response to the reviews, we have made substantial changes, including additional analyses, text, and a major restructuring of the manuscript. Line-by-line responses are below in blue, and line-by-line edits are shown using tracked changes in the revised manuscript document. We believe that this work has been considerably strengthened as a result of your feedback. We would be happy to address any remaining questions and concerns from the editors or reviewers.

Reviewers' Comments:

Reviewer #1 (Remarks to the Author):

The manuscript explores the trends in growing season temperature of several crops (maize, wheat, rice and soybean) as their growing areas move into new environments in response to climate-related temperature changes (adaptive migration). These trends are evaluated against the counterfactual of no adaptive migration.

Overall, the concept of the paper is interesting, and the authors present some new and useful data. However, the current version of the manuscript is confusing for the uninitiated reader, who needs to put considerable effort into understanding the analysis and the conclusions drawn. In other words, clarity could be considerably improved. For instance, introducing several concepts within the same sentence is problematic, inviting misinterpretations and misunderstandings (e.g. Lines 17-23 introduce the concepts of two different types of adaptive responses, namely damage avoidance vs niche filling, as well as in situ adaptation vs migration). I understand the need for brevity, but as Einstein supposedly said: explanations should be as simple as possible, but not simpler!

We agree on this point and have made a number of changes to aid the readability of the manuscript and help to articulate the key concepts clearly.

1. The most substantial change we have made is to move the majority of the discussion about the lower-bound temperature analysis into the supplemental. Much of the confusion in the manuscript (for multiple reviewers) centered on the interpretation of changes to the cold boundary of temperature. Additionally, the magnitude of the lower-boundary temperature changes was small, and the interpretation nuanced. We provide a brief description of our findings in the main manuscript on lines 122-136 and have added additional text to the discussion of these lower-boundary results in a new supplemental section.
2. We point the reviewer to the significantly re-worked introduction. In this section, we have:
 - a. Improved the flow of the first paragraph

- b. Provided further examples about what is meant by an adaptive response and drawn a distinction between that and evolutionary adaptation (lines 27-31).
- c. Taken what was a single sentence explaining and differentiating in-situ adaptation and migration, split it up, and further articulated what is meant by these concepts (lines 31-40).
- d. Further explained the special role that irrigation expansion plays in determining the geographic range of rainfed crops (lines 36-38).

The authors should consider re-writing the abstract by more meaningfully summarising their findings and the likely implications. The current version left me with a very strong ‘so what?’ question and did not entice me reading the rest of the paper.

We have re-worked the abstract to emphasize novelty and importance.

As a general point, I would have expected the authors to be clearer in articulating that shifts in production areas might not always be in response to a changing climate. Rapid urbanisation, competing demands on land from other sectors, changes in demographics and markets can all be contributing factors for crop migration and these effects might be difficult to separate from true climate drivers. While this is acknowledged (lines 140-142 and 152-154), the point should be made earlier and more forcefully.

We agree with this point and now put this idea in the second paragraph of the introduction as well. *“It is important to note that while climate is a central determinant of cropland geography²⁴, many political, demographic, and economic factors influence observed patterns, and therefore the extent of adaptation will be influenced by societal circumstances.”* (lines 39-42).

The analysis appears sound, although I am not an expert in statistics. I have one question, however: given that the slope of differences between counterfactual and experienced temperatures were tested using P-values, I am wondering if the authors did a check for normality of their data, which is a prerequisite for using P-values (Extended Date Figure 2, a-d).

In response to this and other reviewer comments, we have revised our method of p-value calculation. Instead of assuming a normal distribution from standard errors, we now bootstrap our quantile regressions 500 times to find the distribution of the differences in slopes of the counterfactual and observed model. Unlike standard p-values that assume normality, non-parametric bootstrapping allows for confidence intervals to be derived empirically from non-normal data. As before, the differences in slopes are denoted as theta in the manuscript text. P-values for a significant difference between the slopes of the observed and counterfactual models are found as the proportion of data in the distribution of bootstrapped thetas that overlap 0. The following are the resulting histograms of theta for the 95th percentile quantile regressions. As can be seen, the bootstrapped distribution of parameter values does not overlap zero, showing that theta can confidently be distinguished from zero even in the absence of any assumptions of normality.

Data presented in Fig. 4 seems inconsistent in terms of its interpretation. While I agree with the interpretation of the warm bound growing season temperatures (left panels in red), I could not understand why wheat (f) was considered consistent with adaptive migration, while maize and soybeans were not (I agree with the interpretation for rice, panel g). The interpretation appears to be based exclusively on the statistical significance tests as shown in the Extended Data Table 4), but this is not intuitively apparent by visually inspecting the slopes in Fig. 4. For me, this raises the question: Could this simply be an artefact of the statistical analysis? Would a slight change in the significance threshold result in a different interpretation? While the slopes might be statistically significantly different, the magnitude of difference might be irrelevant from a practical perspective (see several papers debating ‘the insignificance of significance testing’).

We believe you are correct. The magnitude of the difference is small and thus possibly not practically relevant. We have moved the lower temperature boundary analysis to the extended data and added a further discussion of the caveats of this analysis to the supplemental. We believe the manuscript has become clearer and more streamlined because of this change.

We include a paragraph in the main text (starting on line 123) that reads:

“The supplemental material includes additional information and analyses on lower bound (5th percentile) temperature changes. The interpretation of these results is more nuanced because it is not entirely clear if moving into warmer areas or even historically cooler areas than the counterfactual is ‘preferable.’ When crops do move into even cooler areas than the counterfactual, they are often still experiencing warmer temperatures than the beginning of the time period due to climate change. Our results show that...”

Specific comments

Lines 2-3 Yet the mechanisms and capacity for agricultural adaptation to climatic changes remain highly uncertain ...

The term ‘uncertain’ does not adequately convey the real issue surrounding agricultural

adaptation. Adaptation is complex and highly context-specific. Some adaptive measures are very certain, but often they cannot be scaled. They need to be seen and evaluated within their socio-economic context. Using the term ‘uncertain’ is probably one of the most unhelpful terms in this context and simply provides fuel for a ‘do nothing’ response (the supposed ‘uncertainty’ of climate science is largely responsible for the lack of policy action so far).

Thank you for this thoughtful comment. The abstract has been re-worked and no longer contains this problematic language. Additionally, some of the ideas on the context-specific and difficult-to-scale nature of many adaptive responses have been acknowledged in lines 180-195.

Lines 13-14 If migration continues into the future, it will have important implications for the magnitude and extent of agricultural climate change impacts ...

While this statement is true, it fails to acknowledge some of the potentially much larger impacts of such migration on e.g. landuse changes or socio-economic circumstances. It is also rather meaningless, because we know a priori that migration will continue to occur.

This is a good point. Our abstract now ends:

“However, while continued migration could limit the exposure of crops to damagingly high temperatures, such adaptive measures may incur substantial costs for farmers or drive environmentally damaging land use change.”

Lines 15-17 ... global yields of wheat, rice, maize, and soybeans are expected to fall by roughly 7%, 6%, 3%, and 3% for every degree Celsius increase in global mean temperature² ...

I find it hard to believe that there is a linear relationship with further temperature increases and yield reductions, even if this is the conclusion by Zhao et al (2017). I haven’t checked the reference and am wondering if this is the correct interpretation of their work? I would expect that as temperatures rise, impacts would become more severe and might even cross thresholds where we could see step-changes in yield patters. I’d like to see some better justification, in addition to this single reference, for this response.

We have changed the opening paragraph of the introduction and added new references; we no longer summarize the Zhao et al. results (which we agree require a longer and more nuanced interpretation).

Lines 17-18 However, agricultural systems will surely respond and actual losses will thus depend on the efficacy of such ‘adaptive’ responses by farmers^{4,8,12} ...

This sentence seems without context. I am not sure which adaptive responses by farmers the authors refer to.

We have re-written the surrounding text, and while this sentence largely remains we think it is now set up better. It is also followed by statements that more thoroughly explain what is meant by adaptive responses and give some specific examples.

Reviewer #2 (Remarks to the Author):

This is an original contribution of broad general interest, that advances understanding of

information of high importance. The approach is novel, and asks a different question than has been previously addressed: to what extent are the growing temperatures of major crops that are actually being experienced moderated by "crop migration" in the face of climate change? The approach is valid and the statistical methods used seem to me generally to be appropriate. The focus on temperature extremes is excellent and is biologically meaningful. My main comments concern the potential for misinterpretation, and also, I found parts of the ms. confusing or difficult to understand.

I am concerned that the narrow scope of the conclusions and interpretation might lead to claims (particularly by news media, public, policy makers) that "everything is fine"--crops are being moved to more favorable areas, so why worry about changes in temperatures affecting yield? First, I would like to see the caveats discussed (more than as brief remarks) that future movement of crops will be constrained in many ways--by urbanization, land ownership, water rights, national boundaries, and other land-use issues. For example, if farmers' lands can no longer sustain productivity of the crops they are growing, they can't just move over to someone else's land; there may be winners in this scenario but also losers. Limits to irrigation are mentioned, but too briefly in my view, and limitations in precipitation (another consequence of climate change) will also impede future maintenance of crop productivity and the ability to migrate crop production. Perhaps most importantly, the impacts of crop migration--current and future--on biodiversity are also staggering. Currently some huge proportion of productive terrestrial land surface is dominated by human use; if and when croplands move, they often are overtaking more and more land occupied by natural communities. The story of the expansion of soy production in South America reported here is also a little known story of the obliteration of the highly biodiverse cerrado, for instance. These issues are not mentioned here and in my view it is important to include that context for the changes reported here.

We agree that the implications regarding biodiversity and land use, as well as limitations of the study deserve further discussion and we regret this initial oversight. We have made a number of changes to the text to address these issues:

1. We put these biodiversity/land use change concerns right up front in the abstract "*...while continued migration could limit the exposure of crops to damagingly high temperatures, such adaptive measures may incur substantial costs for farmers or drive environmentally damaging land use change,*" as well as the opening sentence: "*Climate change is predicted to shift areas of global cropland suitability¹²⁻¹⁴, with potentially important impacts on land use change, biodiversity, socio-economic circumstances, and agricultural productivity,*" We hope that making these statements right away will set a more appropriate tone regarding the serious environmental consequences of agricultural land use change.
2. We have added this substantial paragraph to the end of the manuscript: "*It is important to note that adaptive crop migrations depend on the continued ability to move growing areas and expand irrigation, and the long-term prospects for continued use of these practices are not clear. As the climate changes, areas of crop suitability may shift to the particular detriment of developing countries that tend to be warmer, and further work is needed to ascertain the possibility for adaptive migration as well as the ecological and geopolitical implications of migrating food production. Shifting crop areas may not be a*

sustainable method of adaptation for many reasons. Expanding agriculture into new areas is extremely environmentally damaging, decreasing carbon storage, harming water quality, reducing wildlife habitat and biodiversity⁵²⁻⁵⁵. For example, the expansion of soybean production in South America has had damaging consequences for the highly biodiverse Cerrado biome⁵⁶. The majority of harvested area changes in our dataset appear to result from crop switching (Extended Data Fig. 6) or changes to irrigation (Extended Data Fig. 2), however crop switching can be limited when new technologies or methods are required. Further, increasing irrigation can be problematic as an adaptation strategy when measured against the impact on available water resources or the effect of increased runoff on water quality⁵⁷, and the long-term sustainability of irrigation expansion is not clear given existing stress on water supplies⁴⁹⁻⁵¹.

There were a number of areas I found confusing. I don't understand what thermal niche refilling means; this could be better explained.

We agree that “thermal niche re-filling” along with some of the other concepts surrounding the lower boundary temperature analysis were confusing. Please see the response to your next comment regarding how we have dealt with this.

Additionally, we have made an effort to further explain some concepts in the introduction that we believe may have been confusing, including “adaptation,” “in-situ adaptation,” and “migration,” (lines 27-36) We also now include additional text explaining the role of irrigation in our analysis. We hope this has clarified some of the confusing ideas (lines 36-39).

The explanations for the cool bounds were confusing (lines 57-64 and lines 100-107); these portions of the ms. were just hard to understand. The cool season boundary may be less meaningful than growing degree days (is the cool season boundary reducing growing season length?). Since growing seasons were apparently calculated, it would be valuable to include those results for interpreting the findings.

In response to this and other reviewer comments, the lower temperature bounds analyses are mentioned briefly, related figures are moved to extended data, and the discussion about this is moved to a new supplemental section. The supplemental text is also expanded to include further explanations and caveats that we hope improve clarity and interpretation of the cool bound analysis for readers. We hope that this change improves the focus of the manuscript while still presenting the lower boundary analyses for those interested enough to explore the supplementary material.

We have added a paragraph to the main text that briefly describes the lower boundary results (lines 123-137).

“The supplemental material includes additional information and analyses on lower bound (5th percentile) temperature changes. The interpretation of these results is quite a bit more nuanced because it's not entirely clear if moving into warmer areas or even cooler areas than the

counterfactual is ‘preferable’, and when crops do move into even cooler areas than the counterfactual they are often still experiencing warmer temperatures than the beginning of the time period due to climate change. In addition, it is not as clear that cold temperatures are as serious a constraint on production as hot temperatures⁴, meaning the adaptive response to hotter temperatures would be expected to be less pronounced. In our dataset, maize, wheat, and soybean growing in the coldest 5% of their range have lower yields (by about 16% on average) than the middle 90% of areas, but rice actually has yields that are about 6% higher in the coldest area (Extended Data Fig. 5). The magnitude of change between the counterfactual and observed models was relatively small for all crops. We found that maize, rice, and soybean experienced 5th percentile growing season temperature increases that were 0.14, 0.49, and 0.22 °C less than they would have been in a counterfactual situation, while wheat experienced increases in growing season temperature that was 0.23 °C greater than the counterfactual.”

Are wheat and rice contracting overall or just changing to irrigation (Fig 3)? Is wheat being replaced by maize, or is it just that rainfed wheat is being replaced by irrigated wheat? Why is wheat shifting out of the coldest areas in Canada and Russia? The expansion of irrigation is mentioned but could be somewhat expanded; this is an important part of the story.

We agree that we could do more to visualize and discuss changes in irrigation and crop switching. We now include a map of changes to irrigated harvested area (Extended Data Fig. 2) as well as a map of the largest expansions and contractions in each grid cell (Extended Data Fig. 3). An expanded discussion of these issues can be found on lines 85-104:

“To understand how changes in crop areas may have moderated exposure to warming temperatures, we then map trends in rainfed harvested areas of each crop between 1973 and 2012, with increases shown in green and decreases in brown (Fig. 3). Total rainfed and irrigated areas together increased to varying degrees for each crop over this time period (+35% maize, +0.3% wheat, +13% rice, and 159% soybean), however, because adding irrigation decreases rainfed areas, total rainfed areas for wheat and rice decreased by 10% and 7%, respectively. Rainfed maize areas increased by 24% (compared to the 35% increase in total area), and rainfed soybean areas increased by 158% (the majority of increases in soybean areas were rainfed). For cross-referencing, maps of changes to irrigated harvested area for each crop are presented in Extended Data Fig. 2 ... It is not possible to determine exactly if one crop is being replaced with another, in part because we do not track crops outside of the four presented here. For example, the contraction of wheat in Canada and Russia may be linked to the expansion of rapeseed production^{6,45}, however, we are unable to show that directly. We do provide categorical maps of the largest expansions and contractions of the four crops analyzed here in Extended Data Fig. 3.”

Figure 1 was useful, but somewhat unclear; what are the axes? Are the boxes meant to show total area? Why is it different in c and d compared to a? What does the shading indicate?

We have edited Fig. 1 and the Fig. 1 caption to clarify these issues. The portions of Fig. 1 related to the initial timestep are now labeled t_1 , and those related to a later timestep are now labeled t_2 .

Additionally, we have added the following text to the Fig. 1 caption: “*a-e represent theoretical gridded maps of crop harvested area. Dark green grid cells have the largest fraction of harvested area, decreasing as the shade gets lighter. Map ‘a’ represents the initial time period, while maps ‘b-e’ represent theoretical scenarios at a later time.*”

In Fig 2 a-d does it make sense to show the average growing season temperature, if it is changing rapidly? Maybe the baseline year and the slope for change over time would make more sense?

We agree this makes more sense for consistency with the rest of the analysis. We have changed a-d to visualize growing season temperature at the beginning of the time series.

I don’t understand the point of extended figure 2 (why is this useful information? What does it tell us?); spatially explicit values of fit and significance seem, well, kind of odd. Maybe there's something there I'm missing, so please explain if so, or delete this figure.

The linear regressions of change in growing season temperature presented in the main text Fig. 2 are done at the grid cell level. Consequently, the associated p-values and R-squared values differ spatially. The reason to include this figure is to identify areas where we should be cautious about interpreting the importance of temperature trends presented in Fig. 2.

Extended data Fig 1 could be explained better; the caption and ms. text are unclear.

The figure and caption have been revised to improve clarity (note that this is now Extended Data Fig. 5).

Extended figure 3 was just hard to understand (relative change? I found this hard to interpret).

We agree that this figure is difficult to interpret and not entirely necessary. Therefore, we have removed this figure from the supplemental material.

I had a couple of comments on the graphics: The colors in Fig 3 would be better using a different palette, because it is showing something very different (the colors in the other figures indicate, appropriately, temperature or temperature-related data; these maps are referring to something completely different). I found the ‘shadows’ on the maps unnecessary--being a fan of Edward Tufte, I would like graphical elements to convey information, not decoration, and the shadows are just a distraction from the data.

We agree that utilizing a different color palette for area changes in Figure 3 is useful to distinguish between area changes and temperature changes. All area changes are now shown in a brown/orange to green color palette. The gray shaded areas on the map do convey information – they show places where crop area is present, but changes in harvested area are zero or near-zero. This is different from the white areas with zero or near-zero area devoted to the crop being mapped. We have added text to the figure caption to make this clearer:

“Locations in brown are experiencing decreases in rainfed harvested areas, locations in green are experiencing increases, and locations in gray are experiencing near-zero trends.”

Jessica Gurevitch

Reviewer #3 (Remarks to the Author):

Review: Climate Adaptation by Crop Migration (Sloat et al. 2019)

1. Summary

In this paper, the authors focus on crop migration as a mechanism for climate change adaptation by looking at the upper (95 percentile) and lower (5 percentile) bounds of average temperature distributions. The analysis focuses on rainfed maize, wheat, rice, and soybeans through the world from 1973-2012 to study how land use for each crop has changed relative to a linear trend in average temperatures. They construct a counterfactual based on 1973-1977 land uses for each crop and compare against the entire sample. The main results suggest crop migration has mitigated most of the damage from warming as opposed to thermal tolerance of crops. These results provide a worldwide impact that extreme heat will play on the distribution of crops and have important implications for food security under climate change.

While the paper attempts to address world-wide crop migration as a mechanism for climate change adaptation the work is similar to Zhu and Tara (2018) and the data used in the analysis could be improved upon (Zhu and Tara provide a similar analysis with better data). Further, it is unclear why crop migration is occurring and the underlying mechanism is not addressed which is a major gap in the current literature. Instead, the authors discuss these possible mechanisms, such as prices, trade, access to markets as limitations to the paper and do not address them in the analysis. Understanding that these mechanisms are difficult to address, the paper does not improve our understanding of crop migration but that it might exist and may mitigate climate change impacts.

We agree with Reviewer # 3's statement that the underlying mechanisms driving land use change and subsequent patterns crop migration, such as prices, trade, access to market etc. are not addressed in our analyses. These mechanisms are important and not well understood (e.g. see Meyfroidt et al. 2018), but are beyond the scope of this particular analysis. Instead, this study identifies that rainfed crop migration is occurring across the globe (never previously shown) and reveals how and why this migration is important as a climate adaptation mechanism. Our insistence on using language consistent with pattern description (e.g. climate migration or niche expansion) over mechanistic hypotheses (e.g. specific policies or genetic evolution) is intended to make clear the scope of what we can and cannot say with these analyses. We believe this analysis represents a substantial contribution to the climate and agriculture field, since crop migration has long been noted as a possible climate adaptation mechanism, but has never been shown for global agricultural systems. Further, we note that there is considerable interest in

migration of species in response to climate change within natural systems, as well as widespread concern over climate change impacts and global food security. As such, we believe our results will be of broad interest to the readers of *Nature Communications*. Further, the focus on a single but important climate variable was also a deliberate choice. As noted by the other reviewers, even with constraining the analysis to temperature, it is still a challenge to concisely communicate our proposed framework for identifying adaptive migration alongside presenting considerable data analyses of high-resolution global datasets. We agree that future research should expand these analyses to better understand what drives changes in geographic patterns of agriculture and the relationship between migration and other climate indices. For now, it is our hope that you find this work sufficiently compelling to stand alone as a clear story of global climate adaptation through crop migration.

We thank Reviewer #3 for pointing out the Zhu and Troy paper, as we found it very useful and now reference it in the manuscript (lines 23 and 24). However, Zhu and Troy do not address adaptation through the movement of crop growing areas, which is the primary focus of our paper and made possible through analysis of our high-resolution, historical global crop area dataset. The Zhu and Troy analysis examines changes in many agriculturally important climate indices under climate change, but they utilize harvested areas that are static over time, meaning they are unable to address questions of climate adaptation through crop migration. The vast majority of scientific literature on climate change and agriculture (both retrospective studies and studies predicting future impacts) do not consider changes in harvested area (e.g. Burke, Lobell, and Guarino 2009, *Global Environmental Change*; Schlenker and Roberts 2009, *PNAS*; Lobell et al. 2011, *Science*; Zhao et al. 2017 *PNAS*; Tigchelaar et al. 2018, *PNAS*; Zhu, Troy, and Devineni 2019, *ERL*), a fact which highlights the uniqueness of our study and our findings on adaptive migration.

2. Major Comments

Good

- Paper is well written and the authors are explicit that their analysis only focuses on crop migration using average temperature distributions. Discussion section addresses the major limitations in the paper and is upfront about what they are doing.
- The discussion of crop migration and changes in crop area distributions sets up the understanding of the analysis nicely.
- Figures and tables are organized and labeled properly. The cropping pattern (Fig 1) is a nice complement to the climate change adaptation discussion.
- Discusses the main papers in the literature but omits one similar to their analysis (Zhu and Tara 2018).

Critic

- The main critic for this review is that the data and modeling strategy could be improved upon. Further, the analysis estimates linear trends when the majority of climate impacts are nonlinear (second-order direct and indirect effects).

We have improved our analyses in several ways following review, including additional analyses with second-order responses. More details follow.

◦ Data:

- Why only use average temperatures? This is a midpoint which does not account for the entire distribution of temperatures during the day or growing season. Degree days would be a better measure and likely more defensible given extreme degrees are known to be strong predictors in yield models.
- Linear interpolation of daily average temperatures using monthly values is problematic. Instead of linear interpolation of monthly data, I would suggest using a reanalysis gridded data set. There is better data available for the analysis that would improve the confidence in the results.

We utilize the widely-used CRU monthly dataset because other global gridded daily temperature datasets are either (1) coarser resolution (HadGHCND), (2) shorter duration (ESRL CPC) or (3) still in beta testing (Berkeley Earth). We agree that reanalysis datasets provide temperature fields at high temporal resolution, but there are known biases in reanalysis surface temperatures that can be considerable in managed landscapes (e.g. Kalnay and Cai 2003 *Nature*, and the literature on “observation minus reanalysis”). Some forcing datasets (e.g. Princeton Global Forcing, AgMERRA) correct reanalysis biases using observational temperature, but we prefer to work directly with observational datasets, circumventing the need for correction.

Despite the shorter duration of the daily ESRL CPC, we have comprehensively analyzed the sensitivity of our results to using this daily data. First, we provide a comparison of MDT calculated from CRU as used in our paper with MDT calculated in the same way using data from ESRL CPC from the time period in common between both datasets, 1979-2012. The following graph is for a single grid cell in the midwestern US. The orange line is the CRU data (interpolated between months), while the blue line is the ESRL CPC (daily resolution). Because mean daily temperatures are averaged over the growing season in our analysis, you can visually estimate that the values are similar.

To further quantitatively examine this relationship across the global range of each crop, we provide the results of a regression between MDT calculated from the ESRL CPC data (y-axis) and the CRU data (x-axis). Adjusted R^2 values are above 0.95 for all crops. Additionally,

Burke, Hsiang and Miguel (2015) show that under a reasonable set of assumptions, a non-parametric yield response based on daily temperature distributions (of which degree day models are one version) can be parameterized with a smooth function of average annual temperatures, reinforcing the validity of examining average annual growing season temperatures as we do here.

To be sure that the daily temperature records lead to the same conclusion as utilizing monthly data, we repeat our entire adaptation analysis for all crops using the ESRL CPC data and include the results in Extended Data Tables 6 and 7. We find the conclusions using ESRL CPC data are essentially the same as those presented in the paper using CRU data. That is, the differences between the counterfactual and observed 95th percentile quantile regression slopes are of similar magnitude, direction, and significance.

Note: In re-analyzing our data and comparing it to other temperature datasets we discovered an error in our code that effected a small number of grid cells. Consequently, you will see minor updates to some of the numbers in the paper, although the changes are small in magnitude and none of the conclusions have changed.

- What is the climate variable? Average daily temperatures? The main variable accounts for daily average temperatures which are a measure of weather, not climate. A better variable would be previous 30-year average temperatures or a rolling mean of temperatures under different windows (e.g. 30 years).

This analysis concerns the distribution of global growing season average temperatures. The 95th percentile quantile regression focuses the analysis on the hottest, most marginal areas where yields are likely to suffer. The benefit of annually-dynamic regressions over 40-years is that there is no arbitrary cut off to evaluate temperature exposure before or after a given time.

◦ Modeling strategy:

▪ Quantile regressions make sense to capture changes on the ends of the distribution, but I would suggest using nonlinear quantile regression. While the linear trends do capture average temperatures for most crops it is not always the case (see wheat). I suspect your results will change with different percentile bounds and climate variables.

In order to explore the possibility of non-linear fits, we added a non-linear response function (a second-degree polynomial) to our quantile regressions for all crops. This term is significant (p-value < 0.05) for three of the eight models (Extended Data Table 4). The significant non-linear fits to wheat (counterfactual) and soybean (both observed and counterfactual) are visualized in Extended Data Fig. 4. However, the addition of this term did not appear to impact the general direction or overall conclusions drawn from our results. Therefore, the more easily interpreted linear results are the ones presented in the main text of the paper, while the non-linear analyses are mentioned in the methods section under ‘Linear regressions of temperatures and harvested areas’.

▪ How sensitive are the bounds? I would suggest showing how results change with different percentiles (90%, 99%, etc.)

We now include additional quantile regression analysis results at additional percentiles, including 90th, 93rd, 97th, and 99th percentiles. These results are presented in Extended Data Table 2, Extended Data Table 3, and are mentioned in the text on lines 117-120. Briefly, results for our adaptation test at all alternative warm bound thresholds (90th, 93rd, 97th, and 99th percentiles) remain consistent with our findings for the 95th percentile across all crops, except the 99th percentile (all crops) and the 90th percentile (just maize). These exceptions support results in the same direction as the 95th percentile but have P-values > 0.05. Insignificant P-values at the 99th percentile are likely due to standard statistical fluctuations in extreme values at this percentile.

▪ The data used in the analysis is based on linear interpolation or a linear variable (average temperatures) which is then modeled using a linear model. An analysis using nonlinear assumptions would improve confidence in the results.

These are good points. Please see the above responses on our daily interpolation versus native daily data as well as the additional analysis using nonlinear temperature trends. We hope that our efforts have adequately addressed these concerns.

▪ While on average temperatures are increasing what about regionally? I would suggest adding a region fixed effect to account for the fact that not all areas will increase uniformly. A break out of these regions and increases would be another figure to look at and see if there is a significant difference.

Spatial patterns of growing season temperature trends for all 5 arc-minute grid cells are shown in Figure 2. As we are interested in capturing trends in extreme growing conditions across all crop areas, we analyze the entire dataset using quantile regression and do not use any fixed effects by

location. However, we agree it is useful to summarize trends regionally and have added a regional summary to Extended Data Table 5.

3. Minor Comments

- Line 19-23 is an important sentence but the sentence is long and clunky. I would suggest breaking it up.

Done

- Line 36: Defend 5% and 95% percentiles.

We focus on the 95th percentile as our default warm bound, but, as noted earlier, we have added additional percentiles in the supplemental. Although the choice of any particular percentile value is somewhat arbitrary, even for the warm bound (e.g. 93th vs. 95th vs. 97th), our supplemental analyses demonstrate the robustness of the main results presented using the 95th percentile. As mentioned earlier, we have moved the cool bound analyses to the supplemental materials.

- Suggestion: A plot of temperature distributions across yields and how the 5% and 95% percentile distribution differ to actual vs counterfactual.
- Suggestion: a plot of how much yield is located on those upper and lower bounds. Also, look at changes in actual vs counterfactual.

We agree that the suggested plots would be interesting, however, our dataset does not differentiate between rainfed and irrigated yields. In order to compare yields between different temperature percentiles we obtained a second dataset for which rainfed and irrigated yields and harvested areas were estimated through data synthesis and simple hydrologic modeling (Siebert, S. & Doll, P. Quantifying blue and green virtual water contents in global crop production as well as potential production losses without irrigation. *J. Hydrol.* **384**, 198–217 (2010)). However, these yields are only relevant circa 2000 and, therefore, comparisons between the observed and counterfactual models are not possible. We do include a plot of rainfed yields circa 2000 at the upper and lower bounds of temperature and those results are presented in Extended Data Figure 5.

- Line 57: “Changes in the cooler bound ... will make areas at this bound more favorable.” Is this true for all crops? Increases in temperatures at the lower bound for wheat might be less favorable depending on location. For example, wheat varieties favor winters that rely on frost levels for production.

We agree with the reviewer that the interpretation may be nuanced with regard to winter season crops. This paragraph was moved to supplementary information and the statement “...increases in temperatures will make areas at this bound more favorable” was changed to “... increases in *growing season* temperatures *may* make areas at this bound more favorable.”

- Line 90: Why 1973-1977 as the counterfactual? Is this a stable time period to conduct a

counterfactual with? A discussion as to why this interval was used and how sensitive the results are to changing these dates (e.g. 1973-1980).

We want to compare dynamic harvested areas to harvested areas at the beginning of the time period. We are interested in the average initial conditions, so instead of simply choosing the first year we averaged over the first five years. This helps average out the annual variability in harvested areas that can occur for a variety of reasons (fallow patterns, price shocks, etc.).

- Line 139: are there implications for complementarities, such as corn and soybean? How has warming in soybean areas related to corn?

We have now added maps of the biggest expansions and contractions in terms of rainfed harvested area (Extended Data Fig. 3), as well as some further explanation to the main text: *“It is not possible to determine exactly if one crop is being replaced with another, in part because we lack data on growing areas beyond the four major field crops presented here. For example, the contraction of wheat from Canada and Russia may be linked to the expansion of rapeseed production, however, our analyses would not show that. We do provide categorical maps of the biggest “winners” and “losers” among the four crops analyzed here in Extended Data Fig. 3.”*

- Suggestion: a plot of changes in trend in different countries/regions.

We have added a regional summary of changes in harvested area and changes in mean temperatures to Extended Data Table 5.

References

Zhu, Xiao, and Tara J. Troy. "Agriculturally Relevant Climate Extremes and Their Trends in the World's Major Growing Regions." *Earth's Future* 6, no. 4 (2018): 656-672.

Meyfroidt, Patrick, et al. "Middle-range theories of land system change." *Global environmental change* 53 (2018): 52-67.

Burke, Marshall, Solomon M. Hsiang, and Edward Miguel. "Global non-linear effect of temperature on economic production." *Nature* 527.7577 (2015): 235.

REVIEWERS' COMMENTS:

Reviewer #1 (Remarks to the Author):

The manuscript has improved considerably following the review. The authors have taken the reviewers' comments into account and made substantive changes in terms of

- clarity (the language is much more precise, and the manuscript is much easier to read)
- data analysis (e.g. using non-parametric bootstrapping to empirically derive p-values)
- data representation (the graphical representation of the data has improved considerably) and
- data interpretation (e.g. by moving the somewhat problematic, lower temperature boundary analysis to the extended data section).

I still feel that the abstract misses an important, final statement that makes it clear that while climatic changes might provide an impetus for crop migration, it might not be the actual driver. An example of such a concluding statement could be:

"Whether crop migration continues will depend on country-specific, socio-economic conditions and policy settings as well as land availability, land suitability and climate drivers."

Following some careful technical editing, I recommend that the manuscript be published.

Professor Holger Meinke
Strategic Research Professor - Global Food Sustainability
University of Tasmania
Australia

Reviewer #2 (Remarks to the Author):

I am largely satisfied with the changes made in response to the comments. The two things I would still like to see, as requested in my original comments, are the change in growing season length and better explanation of the thermal niche results. I believe that you should have this information available from your analyses already, or it should be possible to calculate it without great difficulty. This is an interesting and valuable study with a nuanced and creative approach, and will make an excellent contribution to the literature on biotic responses to climate change. I appreciate the changes about the lower temperature bounds, but the thermal niche explanation is still somewhat obscure.

Reviewer #3 (Remarks to the Author):

Review #2 for "Climate adaptation by crop migration."

The revised manuscript has addressed the majority of my concerns with additional clarification and analyses. Further, the structure and clarity of the paper have greatly improved. I agree that the manuscript is of broad interest to the readers of Nature Communications and provides a new direction for research endeavors, such as species migration and food security. My main concern for understanding why crop migration has shifted remain, but hope this paper will help push research into a direction to understand this important adaptation mechanism.

Minor Comments:

Line 27 - "Avoid" is too strong of a word here. Adaptation does not always avoid damages, but instead mitigates damages. I would suggest "mitigate" instead of "avoid."

Line 30 - "environment" should be "environments"

A. John Woodill

REVIEWERS' COMMENTS:

Reviewer #1 (Remarks to the Author):

The manuscript has improved considerably following the review. The authors have taken the reviewers' comments into account and made substantive changes in terms of

- clarity (the language is much more precise, and the manuscript is much easier to read)
- data analysis (e.g. using non-parametric bootstrapping to empirically derive p-values)
- data representation (the graphical representation of the data has improved considerably) and
- data interpretation (e.g. by moving the somewhat problematic, lower temperature boundary analysis to the extended data section).

I still feel that the abstract misses an important, final statement that makes it clear that while climatic changes might provide an impetus for crop migration, it might not be the actual driver. An example of such a concluding statement could be:

“Whether crop migration continues will depend on country-specific, socio-economic conditions and policy settings as well as land availability, land suitability and climate drivers.”

Following some careful technical editing, I recommend that the manuscript be published.

The length of the abstract needed to be substantially reduced but I was able to end the abstract with a statement that reflects the essence of this comment: “However, continued migration may incur substantial environmental costs and will depend on socio-economic and political factors in addition to land suitability and climate.”

Professor Holger Meinke
Strategic Research Professor - Global Food Sustainability
University of Tasmania
Australia

Reviewer #2 (Remarks to the Author):

I am largely satisfied with the changes made in response to the comments. The two things I would still like to see, as requested in my original comments, are the change in growing season length and better explanation of the thermal niche results. I believe that you should have this information available from your analyses already, or it should be possible to calculate it without great difficulty. This is an interesting and valuable study with a nuanced and creative approach, and will make an excellent contribution to the literature on biotic responses to climate change. I appreciate the changes about the lower temperature bounds, but the thermal niche explanation is still somewhat obscure.

- Additional explanation was added regarding thermal niche expansion on lines 98-99.

- Our crop-specific growing season lengths were not potential growing season lengths based on climate. We utilized a widely-used, global dataset on crop growing seasons that is based on empirical planting and harvesting dates that are static in time (Sacks et al. 2010). We know of no such data that would allow for annually dynamic planting and harvesting dates for all four crops at a global scale.

Reviewer #3 (Remarks to the Author):

Review #2 for "Climate adaptation by crop migration."

The revised manuscript has addressed the majority of my concerns with additional clarification and analyses. Further, the structure and clarity of the paper have greatly improved. I agree that the manuscript is of broad interest to the readers of Nature Communications and provides a new direction for research endeavors, such as species migration and food security. My main concern for understanding why crop migration has shifted remain, but hope this paper will help push research into a direction to understand this important adaptation mechanism.

Minor Comments:

Line 27 - "Avoid" is too strong of a word here. Adaptation does not always avoid damages, but instead mitigates damages. I would suggest "mitigate" instead of "avoid."

changed

Line 30 - "environment" should be "environments"

changed